# The Combined Use of Copper Sulfate and Trichlorfon Exerts Stronger Toxicity on the Liver of Zebrafish

**DOI:** 10.3390/ijms241311203

**Published:** 2023-07-07

**Authors:** Jianlu Zhang, Mingzhen Zhu, Qijun Wang, Hui Yang

**Affiliations:** 1Shaanxi Key Laboratory of Qinling Ecological Security, Shaanxi Institute of Zoology, Xi’an 710032, China; 2College of Urban and Environmental Sciences, Northwest University, Xi’an 710127, China; 3College of Animal Science and Technology, Yangzhou University, Yangzhou 225009, China

**Keywords:** copper sulfate, trichlorfon, combination, transcriptome, enzyme activity

## Abstract

In aquaculture, copper sulphate and trichlorfon are commonly used as disinfectants and insecticide, sometimes in combination. However, improper use can result in biotoxicity and increased ecological risks. The liver plays a crucial role in detoxification, lipid metabolism, nutrient storage, and immune function in fish. Selecting the liver as the main target organ for research helps to gain an in-depth understanding of various aspects of fish physiology, health, and adaptability. In the present study, zebrafish were exposed to Cu (0.5 mg/L) and Tri (0.5 mg/L) alone and in combination for 21 days. The results demonstrate that both Cu and Tri caused hepatocyte structure damage in zebrafish after 21 days of exposure, with the combination showing an even greater toxicity. Additionally, the antioxidant and immune enzyme activities in zebrafish liver were significantly induced on both day 7 and day 21. A transcriptome analysis revealed that Cu and Tri, alone and in combination, impacted various physiological activities differently, including metabolism, growth, and immunity. Overall, Cu and Tri, either individually or in combination, can induce tissue damage by generating oxidative stress in the body, and the longer the exposure duration, the stronger the toxic effects. Moreover, the combined exposure to Cu and Tri exhibits enhanced toxicity. This study provides a theoretical foundation for the combined use of heavy metal disinfectants and other drugs.

## 1. Introduction

Copper (Cu) is an essential trace micronutrient in living organisms that plays a significant role as a co-factor in critical enzyme reactions [1]. Cu has a narrow optimal dose range between essential and toxic levels. When its content exceeds the normal level required for growth and development, it can cause accumulation and irreversible damage to organisms [2]. Many Cu-based pesticides, including cupric sulfate, copper oxychloride, and cupric carbonate, are extensively used worldwide [3]. Copper sulfate is the most commonly used copper preparation in aquaculture due to its strong ability to kill pathogens [4]. It has been widely used to treat bacterial, fungal, and parasitic diseases, and has also been used as an algicide and herbicide [5,6,7,8]. However, excessive Cu can not only kill pathogenic microorganisms, but can also cause damage to fish, such as destruction of the structures of gills, liver, intestine, and other tissues, changes in oxidative stress, and even death [9]. The concentration selection of copper sulfate is a key point in production neutralization research. Some studies choose the concentration of copper sulfate as the concentration unit, while others choose the concentration of Cu^2+^ as the concentration unit. For example, 4 mg/L of (1/10 dose of LC50) copper sulfate was confirmed to have harmful effects on the fertility of male *Nile tilapia* [10], while 0.4 mg/L of Cu^2+^ induced genotoxicity and hepatic impairments in spotted snakehead fish [11]. Meanwhile, studies have demonstrated that a prophylactic bath treatment in copper sulphate has short physical impacts on pikeperch, and the treatment does not significantly impact the health condition of pikeperch [12]. The antioxidant system plays a crucial role in combating oxidative stress and maintaining the redox balance within cells. This system includes important enzymes such as superoxide dismutase (SOD), catalase (CAT), alkaline phosphatase (AKP), and glutathione peroxidase (GPX). These enzymes work together to neutralize reactive oxygen species (ROS) and protect cells from oxidative damage. These components of the antioxidant system contribute to cellular health and play a vital role in various physiological processes.

Trichlorfon (Tri) is one of the organophosphorus pesticides that are widely used in agriculture and aquaculture for insect pest control owing to its relatively low bioaccumulation and short-term persistence [13,14]. Tri exerts its toxic effects on animals mainly by inhibiting acetylcholinesterase (AChE), leading to the accumulation of acetylcholine at nerve synapses and disrupting nerve function [15]. Previous studies have shown that Tri can cause many physiological changes in fish, such as inducing oxidative stress, hematotoxicity, and hepatocyte apoptosis; reducing digestive and absorptive capacity; and disturbing hematological parameters and brain acetylcholinesterase activity [16,17,18].

Copper sulfate is a widely used disinfectant in aquaculture, effectively inhibiting the growth of bacteria, fungi, and planktonic organisms, thus reducing the spread of pathogens. Trichlorfon, also known as an organophosphate insecticide, is a broad-spectrum pesticide that effectively controls pests such as mosquitoes, flies, mites, and copepods in aquaculture systems. The combined use of copper sulfate and trichlorfon allows for simultaneous disinfection and pest control in aquaculture, thereby enhancing the effectiveness of disease and pest management. However, the combined toxic effect of Cu and Tri on fish is still unclear.

In the present study, we use zebrafish as a research model to investigate the combined toxicity of Cu and Tri through histological observation, enzyme activity determination, and transcriptome sequencing. We also explore the individual toxicity of Cu and Tri to zebrafish for comparison. Our results can provide a further understanding of the toxic effects of Cu and Tri on fish and serve as a reference for the safe use of fishery drugs in aquaculture.

## 2. Results

### 2.1. Morphology Observation

The Con group exhibited normal morphology and a tight arrangement of zebrafish hepatocytes (Figure 1A). In the Cu group, there was a significant increase in the gap between hepatocytes (Figure 1B), and the liver tissue morphology of the zebrafish treated with Tri was similar to that of the Cu group (Figure 1C). In the Cu + Tri combined treatment group, the zebrafish hepatocytes exhibited a more scattered arrangement, and some hepatocytes showed a blurred morphology with signs of necrosis (Figure 1D). The transmission electron microscopy (TEM) results also reveal that the hepatocyte structure in the Con group was intact with a normal morphology (Figure 1E). However, in the Cu group, the hepatocytes exhibited the presence of autophagolysosomes along with a significant abundance of vacuoles (Figure 1F). In the Tri group, the hepatocytes exhibited endoplasmic reticulum expansion and the initiation of autophagolysosome formation (Figure 1G). In the Cu + Tri combined treatment group, the presence of autophagolysosomes and endoplasmic reticulum expansion was similarly observed. Additionally, the hepatocytes exhibited a more severe vacuolation phenomenon, accompanied by a significant abundance of vacuolated mitochondria (Figure 1H,I). The relative quantification of the liver cell numbers per unit area of liver tissue is shown in Figure 1J, where the Cu group, Tri group, and Cu + Tri combined group had significantly lower hepatocyte numbers per unit area compared to the control group, at 63.5%, 63.1%, and 38.1% of the control group, respectively. These results indicate that both Cu and Tri can cause liver damage in zebrafish, with a greater combined exposure effect.

### 2.2. Oxidative Stress and Immune Enzyme Activity

As depicted in Figure 2, on the seventh day, the CAT, SOD, and GPX activities were up-regulated significantly by 20.76%, 27.54%, and 124.93%, respectively, while the AKP activity was down-regulated significantly by 18.46% in the Cu group compared to the control group. When administered alone, Tri significantly increased the activities of the SOD, GPX, and AKP by 78.05%, 137.11%, and 38.18%, respectively, whereas the combination of copper sulfate and Tri significantly increased the activities of the CAT, GPX, and AKP by 17.01%, 109.87%, and 72.0%, respectively. On day 21, all three treatment groups exhibited significantly lower CAT, SOD, GPX, and AKP activities compared to the control group.

### 2.3. Transcriptome Analysis

In the present study, transcriptome sequencing was employed to uncover substantial alterations in the expression of zebrafish liver genes induced by copper sulfate and Tri, either alone or in combination. As illustrated in Figure 3, 687 and 668 genes were significantly up-regulated and down-regulated, respectively, in the Cu group, while 714 and 560 genes were up-regulated and down-regulated, respectively, in the Tri group. Similarly, 653 and 693 genes were up-regulated and down-regulated, respectively, in the Cu + Tri group. A Venn analysis revealed that 39 and 41 genes were significantly up-regulated and down-regulated, respectively, across all three treatment groups. Nine DEGs selected at random from the transcriptome findings were subjected to RT-qPCR to validate the RNA-seq outcomes. The results indicate that the changes in these genes were consistent with both the RT-qPCR and RNA-seq, thus indicating the reliability of the transcriptome results (shown in Figure 4). The red region indicates the up-regulation of genes, and the blue region indicates the down-regulation of genes in the heatmap analysis (shown in Figure 5).

### 2.4. GO Enrichment Analysis

As shown in Table 1, in the Cu group, the genes linked to the meiotic cell cycle and cell cycle were significantly up-regulated, while the genes associated with the immune response and cellular response to the cytokine stimulus were significantly down-regulated. For Tri, the genes related to the cell cycle, filament assembly, and apoptotic processes were significantly up-regulated, and the genes involved in the lipid metabolism processes, steroid metabolism, small molecule metabolism, and sterol biosynthesis were significantly down-regulated. The combination of the two significantly up-regulated genes were related to the oxidation–reduction process, response to xenobiotic stimulus, organic hydroxyl compound metabolic process, cellular response to xenobiotic stimuli and alcohol metabolism process, while the significantly down-regulated genes were related to the reproductive process, DNA replication, positive regulation of acrosome reaction, and information recognition.

### 2.5. KEGG Enrichment Analysis

As shown in Table 2, in the Cu group, the pathways related to sugar metabolism, drug metabolism, and steroid hormone biosynthesis were significantly up-regulated, while the pathways related to cell adhesion molecules, IgA production of the intestinal immune network, ribosomes, and cytokine–cytokine receptor interaction were significantly down-regulated. For Tri, the pathways related to carbohydrate metabolism, cofactor biosynthesis, and apoptosis were significantly up-regulated, while the pathways related to steroid biosynthesis, metabolic pathways, N-Glycan biosynthesis, linoleic acid metabolism, and the gut immune network for IgA production were significantly down-regulated. In the combine treatment group, the pathways related to retinol metabolism, linoleic acid, tyrosine metabolism, and drug metabolism were significantly up-regulated, while the pathways related to glycerophospholipid, glycerolipid metabolism, cell cycle, and fatty acid elongation were significantly down-regulated.

## 3. Discussion

Copper sulfate and Tri are commonly utilized in aquaculture due to their potential bactericidal and insecticidal effects [19,20]. However, incorrect and excessive usage can result in significant harm to cultured animals [21]. In this study, we conducted individual and combined toxicity experiments on zebrafish using copper sulfate and Tri. Our findings reveal that copper sulfate and Tri, alone and in combination, causes damage to zebrafish liver tissue and induces changes in the antioxidant enzyme and immune activity of the body tissue. The transcriptome results demonstrate that copper sulfate and Tri, alone and in combination, affect various physiological activities of the body, including cell cycle, body immunity, redox reaction, and metabolic pathways.

Environmental pollutants can trigger the production of excessive reactive oxygen species (ROS) in the body, which can cause peroxide damage [22]. To counter this damage, the body produces an antioxidant enzyme defense system. SOD and CAT are widely present in various body tissues and serve as the first line of defense against oxidative damage. SOD catalyzes the disproportionation of superoxide anion radicals into water and hydrogen peroxide, while CAT converts toxic hydrogen peroxide into nontoxic water. GPX is another important peroxide-decomposing enzyme present in organisms, which catalyzes the conversion of reduced glutathione (GSH) to oxidized glutathione (GSSG). This conversion reduces toxic hydrogen peroxide to non-toxic hydroxyl compounds, thereby protecting biofilm from ROS damage and maintaining normal cell function [23].

In the present study, the inhibited oxidative enzyme activity suggests an excessive generation of reactive oxygen species in the fish body, leading to the occurrence of oxidative stress. In contrast to the antioxidant defense system, the non-specific immune defense system represented by AKP can directly dissolve or digest foreign substances for defense purposes [24]. Many studies have indicated that AKP plays a crucial role in the immune response of fish [25,26]. In this study, Tri alone and copper sulfate + Tri treatment significantly increased the AKP activity in the early stages, and significantly decreased the AKP activity in the later stages of the experiment. This indicates that the liver of the zebrafish activated the non-specific immune defense system due to the stimulation of poison in the early stages, and the AKP activity decreased due to the damage caused by poison in the later stages.

The cell cycle refers to the series of events that occur in a cell, which lead to its division and duplication. However, in this study, both compounds were found to have detrimental effects on zebrafish liver, indicating that the up-regulation of gene expression related to cell cycle processes may be a negative feedback mechanism of the body. Cu^2+^ was shown to significantly affect genes related to immune response processes in Japan scallop and Sydney rock oyster [27,28]. Similarly, in this study, copper sulfate treatment significantly inhibited the expression levels of immune response genes in zebrafish liver, which is consistent with the lower AKP enzyme activity that was observed. Tri was reported to affect the lipid metabolism in allogynogenetic crucian carp, which is consistent with the transcriptome results of this study. The combined toxicity of the two compounds had significant effects on the liver redox process and reproductive process of zebrafish. The production and scavenging of free radicals play a crucial role in the redox process. At 21 days, the activities of the CAT, SOD, and GPX were significantly down-regulated by the combination of the two compounds, and the genes related to the redox reaction process were up-regulated, indicating a negative feedback regulation of the body. Studies have also shown that copper sulfate and Tri treatments significantly affect the reproduction of organisms [29,30]. The transcriptome results in this study demonstrate that both compounds have a significant reproductive toxicity to zebrafish, resulting in a down-regulation of reproductive-related gene expression.

According to the KEGG analysis, the copper sulfate treatment alone significantly decreased the transcription level of the immune-related pathways, which is consistent with the findings from the GO analysis. Studies have shown that Cu^2+^ treatment could induce inflammation and immune response in chicken thymus and induce zebrafish larvae to be sensitive to inflammatory stimuli by inducing the apoptosis of macrophages or neutrophils, resulting in a decreased phagocytic activity of the immune macrophages or neutrophils and an unsustainable immune response [31,32]. In contrast, the Tri treatment alone significantly up-regulated the metabolic pathways related to the body, with the pentose phosphate pathway, fructose and mannose metabolism, and nicotinate and nicotinamide metabolism being the main enriched pathways. Furthermore, the Tri treatment alone significantly up-regulated the apoptosis-related pathways, which is an intrinsic pathway of the cell triggered by any stimulus that causes oxidative stress, mitochondrial disorder, and DNA damage [33]. The combination of copper sulfate and Tri mainly affected the retinol, drug metabolism, linoleic acid, and tyrosine metabolism pathways, and significantly down-regulated the glycerophospholipid metabolism and glyceride metabolism pathways. Glycerophospholipids are the most abundant phospholipids in the human body and are involved in membrane protein recognition and signal transduction, and are associated with various metabolic diseases, such as cardiovascular diseases, hypertension, diabetes, obesity, nervous system diseases, etc. [34,35]. One key difference between copper sulfate and Tri is that Cu^2+^ accumulates in tissues [36], which means that the concentration of Cu^2+^ in zebrafish increases after each water change. The repeated use of copper sulfate in actual production will cause Cu^2+^ to accumulate multiple times in aquatic organisms, resulting in a long-term negative impact on the aquatic environment. With the combined exposure to copper sulfate and Tri, the accumulation of copper sulfate causes initial body damage, which becomes more severe with continuous exposure to stable concentrations of copper sulfate. Although Tri cannot accumulate in the body, its toxicity increases due to the decreased body immunity and antioxidant enzyme activity, resulting in a stronger toxicity when combined with copper sulfate.

## 4. Materials and Methods

### 4.1. Animals and Drug Exposure

The juvenile zebrafish used in this study were 2 months old and of the wild-type AB line. They were bred and raised in a zebrafish recirculating aquaculture system in our laboratory. Healthy and uninjured zebrafish (0.31 ± 0.16 g) were randomly selected and assigned to the experimental glass tanks (approximately 1 L of water/g of fish). The fish were maintained in dechlorinated tap water with a pH of 7.2 ± 0.5, dissolved oxygen at 6.0–7.5 mg·L^−1^ and a temperature of 25 ± 0.5 °C, and were subjected to a 14 h/10 h light/dark cycle. They were fed with chironomid larvae twice daily. Four experimental groups were established, including the control group (Con), copper sulfate group (Cu, Sinopharm, 0.5 mg/L), Tri group (Tri, Jiangshan, 90% purity, 0.5 mg/L) and copper sulfate + Tri group (Cu, 0.5 mg/L + Tri, 0.5 mg/L). Three replicates were conducted for each treatment. The concentrations of copper sulfate and Tri were selected based on the safe concentrations of these two drugs and the actual concentrations used in aquaculture. Half of the water in each tank was replaced every two days with fresh dechlorinated tap water and dosed with the appropriate amount of copper sulfate and Tri to maintain a constant drug concentration.

### 4.2. Analysis of Actual Cu^2+^ and Trichlorfon Concentration in Water Samples

The actual Cu^2+^ concentration and trichlorfon concentration in the water were determined before the addition of copper sulfate and trichlorfon to the culture water (first test) and prior to the water change (second test). Feces and food residues were removed from the water via filtration through a 0.45 μm filter membrane. The concentration of Cu^2+^ in water was determined using inductively coupled plasma mass spectrometry (ICP-MS) (Perkin Elmer, Waltham, MA, USA) in the detection center of Yangzhou University. The concentration of trichlorfon in water was determined using high-performance liquid chromatography (Agilent, Santa Clara, CA, USA), and the results are shown in Figure 6.

### 4.3. Sampling and Morphometry

Fish were euthanized at two time points, 7 and 21 days after exposure. The fish were euthanized using MS-222 (500 mg/L buffered with 200 mg/L NaHCO_3_), drained on filter paper, and then their body weights and lengths were measured. For each treatment, six liver tissues were collected (two livers per tank in triplicate) and fixed using Bouin’s solution for paraffin section analysis. Additionally, nine livers were taken for enzymatic activity (three livers per tank mixed in a sample in triplicate). A total of 21 days after exposure, nine livers were collected for RNA-seq (three livers per tank in triplicate) and were immediately frozen using liquid nitrogen and kept individually at −80 °C; until use. In total, 156 fish were used in this study.

### 4.4. Hepatic Histological Examination

The livers were fixed in Bouin’s solution for 48 h at 4 °C and then wrapped in gauze. They were washed with tap water for 24 h, dehydrated using an ethanol gradient, treated with methyl salicylate for 12−24 h until the tissue was transparent, and finally embedded in paraffin. Using a rotary microtome (Leica, Wetzlar, Germany), 6 μm sections were made and then stained using the hematoxylin eosin (HE) method. Microscopic examination was performed using a CHC binocular microscope (Olympus, Tokyo, Japan). For each hepatic tissue slice, we randomly selected three fields of view to count the number of hepatocytes and calculated the relative quantity of hepatocytes per unit area compared to the control group. For TEM analysis, after being fixed in glutaraldehyde overnight, the hepatic tissue was then washed in phosphate buffer (PBS) and post-fixed in 1% osmium tetroxide. After being dehydrated in a graded series of alcohol and embedded with LR white resin, the blocks were cut using microtome (ULTRACUT, Leica Microsystems Ltd., Wetzlar, Germany). Ultrathin sections, 80 nm thick, were contrasted with uranyl acetate and lead citrate, and examined using a transmission electron microscope (HT7700, Hitachi, Tokyo, Japan).

### 4.5. Enzyme Activity Assay

To assay the activity of catalase (CAT), superoxide dismutase (SOD), glutathione peroxidase (GPX), and alkaline phosphatase (AKP), a 9-fold volume of pre-cooled phosphate-buffered solution (PBS) (1 g of tissue: 9 mL PBS) was added to tube with the livers. The liver tissues were mechanically homogenized in an ice water bath and then centrifuged at a speed of 4000 rpm for 10 min at 4 °C. The supernatant was isolated for enzyme activity analysis according to the manufacturer’s protocol of the relevant kit (Nanjing Jiancheng, Nanjing, China). The final activities of the enzymes were normalized to protein concentrations of the corresponding samples. The total protein concentrations of the samples were determined using a protein quantification kit (Beyotime, Shanghai, China).

### 4.6. Transcriptomic Analyses

Total hepatic RNAs were extracted using Trizol (Takara, San Jose, CA, USA) following the manufacturer’s instructions. The genomic DNA was removed using DNase I. The mRNA was then checked for quality, and Library Preparation and Sequencing were conducted as in previous studies [37]. Differential expression analysis was performed using the DESeq2 software (v1.62) to identify the differentially expressed genes (DEGs) between treatment groups and controls. Genes with absolute fold change ≥2, false discovery rate (FDR) < 5%, and average read count in at least one sample ≥10 were considered as DEGs. All DEGs for each group were used to map and identify enriched Gene Ontology (GO) terms (http://www.geneontology.org/ (accessed on 22 July 2021)). Pathway enrichment analysis was performed using the KEGG pathway database. FDRs at the threshold below 0.05 (an adjusted *p*-value < 0.05) were designated as significantly enriched GO terms and pathways in DEGs. Gene co-expression networks were built according to the normalized expression values of genes selected from genes in significant GO terms and pathway terms. For each pair of genes, we calculated the Pearson correlation and chose the significant correlation pairs (FDR < 0.05) to construct the network. Within the network analysis, degree centrality is the simplest and most important measure of the centrality of a gene within a network that determines the relative importance. Degree centrality is defined as the link numbers that one node has to the other. The purpose of network structure analysis is to locate core regulatory factors (genes) in one network. Core regulatory factors connect most adjacent genes and have the biggest degrees.

### 4.7. qPCR Verification of DEGs

Nine genes were selected randomly to validate the RNA-seq data using qRT-PCR assays with SYBR qPCR Mix (Vazyme Biotech Co., Ltd., Nanjing, China) and qPCR (ABI) assays. The reference genes used in this study were beta actin (*actβ*) and eukaryotic translation elongation factor 1 alpha (*ef1α*). The relative transcript changes were calculated using 2^−ΔΔCq^ method [38], and the arithmetic mean of the two Cq values of the reference genes was used to determine the relative transcript changes. The primer sequences are listed in Table 3. The qPCR efficiency for all genes ranged from 90% to 110%.

### 4.8. Statistical Analysis

All statistical data were reported as mean ± standard deviation (SD). Normality of distribution was tested using the Kolmogorow–Smirnoff test, and homogeneity of variance was checked using Levene’s test. One-way analysis of variance (ANOVA) was conducted to analyze the data, followed by Tukey’s post hoc test for pairwise comparisons between groups with significant differences. A *p*-value of less than 0.05 was considered statistically significant. Statistical analysis was performed using SPSS version 18.0.

## 5. Conclusions

In this study, zebrafish were exposed to copper sulfate and Tri alone and in combination. The results reveal varying degrees of damage inflicted on zebrafish liver tissue by copper sulfate and Tri alone, and the damage was more pronounced when these two compounds were combined. Additionally, significant changes were observed in the activities of antioxidant enzymes (SOD, CAT, GPX) and immune enzymes (AKP). The transcriptome analysis showed distinct alterations in the genes and pathways associated with multiple life processes in response to copper sulfate and Tri exposure, whether alone or in combination.

## Figures and Tables

**Figure 1 ijms-24-11203-f001:**
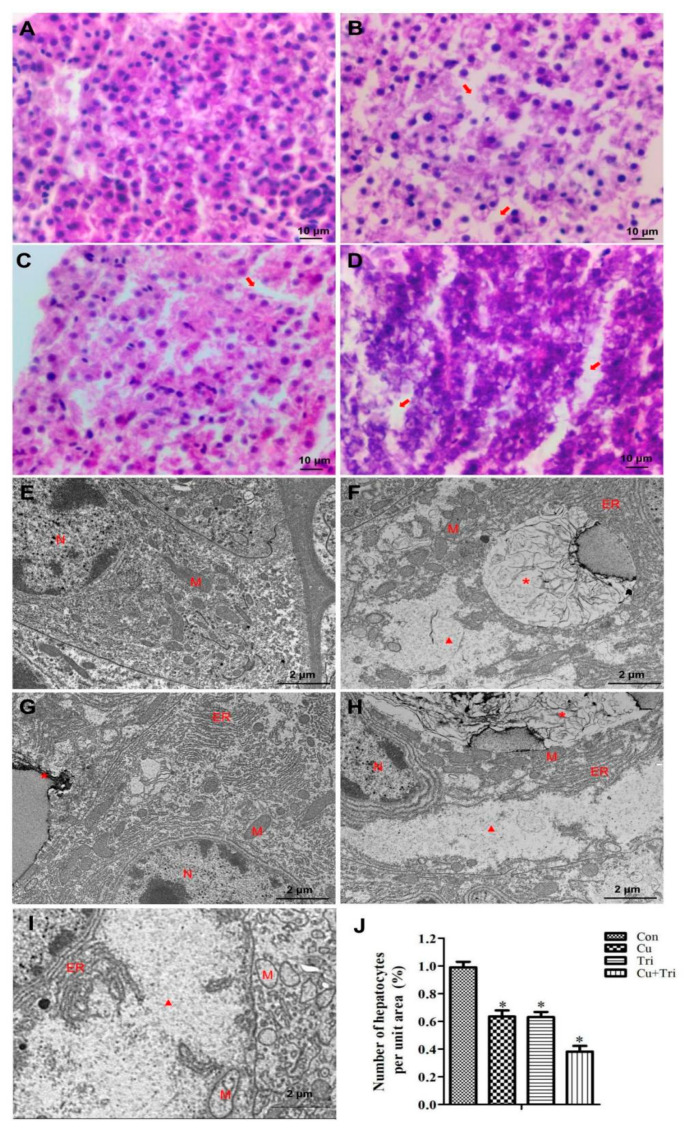
Effect of copper sulfate and trichlorfon on the liver of zebrafish on the 21st day. (**A**–**D**) Liver tissue paraffin sections; (**A**) Con; (**B**) Cu; (**C**) Tri; (**D**) Cu + Tri. (**E**–**H**) Liver tissue transparent electron microscopy; (**E**) Con; (**F**) Cu; (**G**) Tri; (**H**,**I**) Cu + Tri. (**J**) Relative number of hepatocytes per unit area of the hepatic slice. Bars indicate mean ± SD (n = 3). * indicates *p* < 0.05. The arrow indicates the gap between hepatocytes. * indicates autophagolysosome; triangle indicates intracellular vacuoles; N: nucleus; M: mitochondria; ER: endoplasmic reticulum.

**Figure 2 ijms-24-11203-f002:**
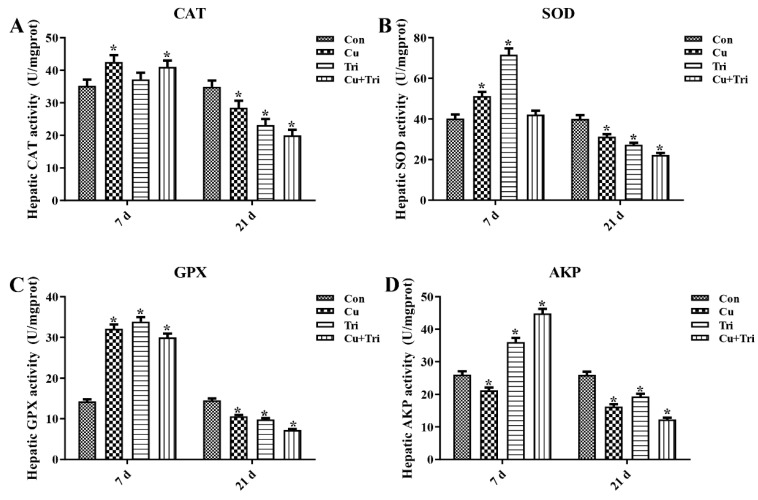
Effect of copper sulfate and trichlorfon on antioxidant and immune enzyme activity. (**A**) Catalase (CAT); (**B**) superoxide dismutase (SOD); (**C**) glutathione peroxidase (GPX); (**D**) alkaline phosphatase (AKP). Con is the control group. Cu is the copper sulfate exposure group. Tri is the trichlorfon exposure group. Cu + Tri is the combination exposure group. Bars indicate mean ± SD (n = 3). * indicates *p* < 0.05. The term “statistical significance” indicates that there is a significant difference when comparing each group of data to the control group.

**Figure 3 ijms-24-11203-f003:**
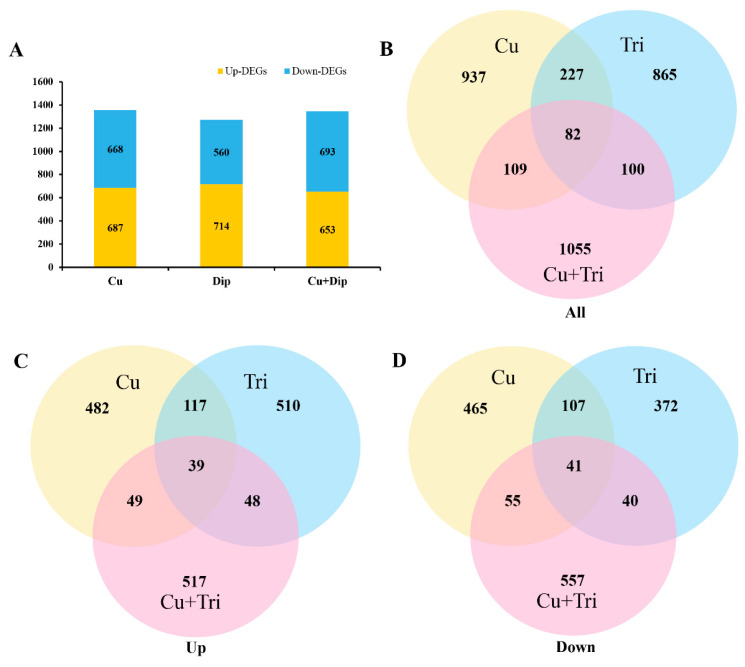
(**A**) Transcriptome sequencing analysis of number of DEGs in three treatment groups compared to the control. (**B**–**D**) Venn analysis for DEGs.

**Figure 4 ijms-24-11203-f004:**
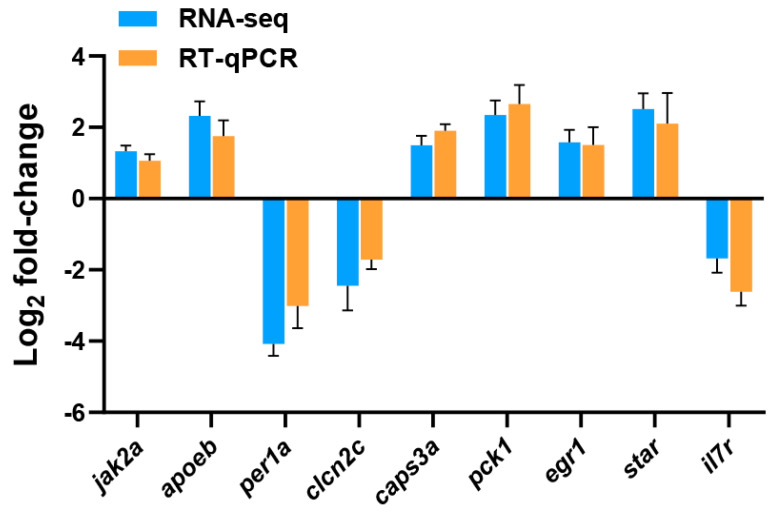
RT-qPCR verification of nine DEGs. *ef1α* and *actβ* were used as the reference genes; bars indicate mean ± SE (n = 3).

**Figure 5 ijms-24-11203-f005:**
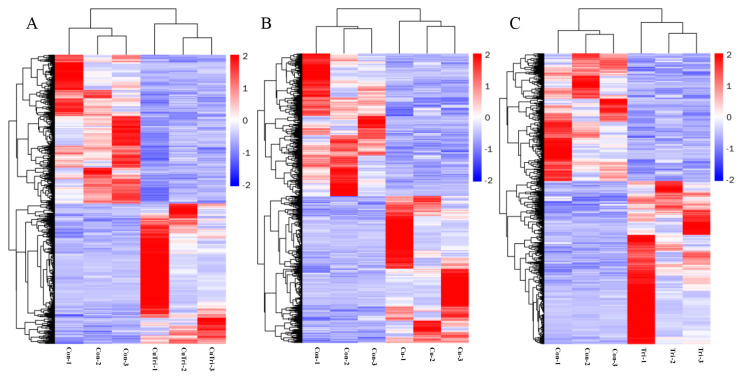
Heatmap analysis (columns represent samples, and rows represent genes. The red region indicates up-regulation of genes, and the blue region indicates down-regulation of genes). (**A**) Comparison between control and copper sulfate + trichlorfon group. (**B**) Comparison between control and copper sulfate group. (**C**) Comparison between control and trichlorfon group.

**Figure 6 ijms-24-11203-f006:**
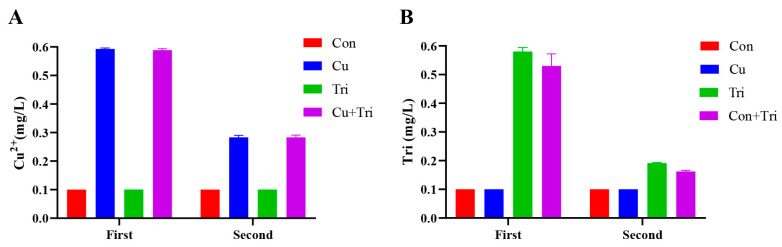
Actual Cu^2+^ and trichlorfon concentration in water samples. (**A**) represents the concentration of copper ions. (**B**) represents the concentration of trichlorfon. The first measurement refers to the initial determination of copper sulfate and trichlorfon concentrations immediately after adding them to the water. The second measurement refers to the determination of concentrations prior to changing the water for the experiment.

**Table 1 ijms-24-11203-t001:** Top ten significantly enriched GO terms including up and down.

		GO_ID	Term	Test	Ref.	*p*-Value
**Cu**	**Up**	GO:0051321	Meiotic cell cycle	12	101	1.71 × 10^−7^
		GO:0006030	Chitin metabolic process	5	10	3.45 × 10^−7^
		GO:0007049	Cell cycle	39	1030	3.72 × 10^−6^
		GO:0015793	Glycerol transport	4	9	1.02 × 10^−5^
		GO:0034508	Centromere complex assembly	5	19	1.40 × 10^−5^
	**Down**	GO:0006955	Immune response	37	573	3.24 × 10^−12^
		GO:0060326	Cell chemotaxis	15	152	4.73 × 10^−8^
		GO:0019882	Antigen processing and presentation	8	34	7.51 × 10^−8^
		GO:0050900	Leukocyte migration	13	135	4.97 × 10^−7^
		GO:0071345	Cellular response to cytokine stimulus	16	219	1.06 × 10^−6^
**Tri**	**Up**	GO:0035082	Axoneme assembly	7	38	5.14 × 10^−6^
		GO:0007049	Cell cycle	40	1030	7.18 × 10^−6^
		GO:0019755	One-carbon compound transport	5	16	7.50 × 10^−6^
		GO:0042981	Regulation of apoptotic process	27	579	1.04 × 10^−5^
		GO:0001578	Microtubule bundle formation	7	47	2.22 × 10^−5^
	**Down**	GO:0006629	Lipid metabolic process	22	710	5.38 × 10^−6^
		GO:0016126	Sterol biosynthetic process	5	30	1.33 × 10^−5^
		GO:0008610	Lipid biosynthetic process	13	294	1.34 × 10^−5^
		GO:0008202	Steroid metabolic process	7	85	2.94 × 10^−5^
		GO:0044281	Small molecule metabolic process	27	1144	6.11 × 10^−5^
**CuTri**	**Up**	GO:0055114	Oxidation–reduction process	40	880	7.50 × 10^−8^
		GO:0009410	Response to xenobiotic stimulus	10	101	1.37 × 10^−5^
		GO:1901615	Organic hydroxy compound metabolic process	14	203	1.91 × 10^−5^
		GO:0071466	Cellular response to xenobiotic stimulus	9	86	2.36 × 10^−5^
		GO:0006066	Alcohol metabolic process	12	157	2.73 × 10^−5^
	**Down**	GO:0022414	Reproductive process	21	252	9.57 × 10^−9^
		GO:0006260	DNA replication	14	119	4.05 × 10^−8^
		GO:0060046	Regulation of acrosome reaction	6	14	9.75 × 10^−8^
		GO:2000344	Positive regulation of acrosome reaction	6	14	9.75 × 10^−8^
		GO:0009988	Cell–cell recognition	6	15	1.60 × 10^−7^

**Table 2 ijms-24-11203-t002:** Top ten significantly enriched KEGG pathways including up and down.

		ID	Description	Test	Ref.	*p*-Value
**Cu**	**Up**	dre00520	Amino sugar and nucleotide sugar metabolism	7	63	5.35 × 10^−5^
		dre00140	Steroid hormone biosynthesis	6	67	0.000613967
		dre00983	Drug metabolism—other enzymes	7	87	0.000414833
		dre00980	Metabolism of xenobiotics by cytochrome P450	5	61	0.002632369
		dre00040	Pentose and glucuronate interconversions	4	42	0.004145541
	**Down**	dre05168	Herpes simplex virus 1 infection	11	185	0.000308054
		dre04514	Cell adhesion molecules	8	152	0.004449689
		dre04672	Intestinal immune network for IgA production	4	37	0.00351935
		dre03010	Ribosome	6	130	0.023987246
		dre04060	Cytokine–cytokine receptor interaction	8	204	0.023658627
**Tri**	**Up**	dre00030	Pentose phosphate pathway	3	33	0.018950865
		dre00051	Fructose and mannose metabolism	3	45	0.04248255
		dre00760	Nicotinate and nicotinamide metabolism	3	45	0.04248255
		dre01240	Biosynthesis of cofactors	7	188	0.044177541
		dre04210	Apoptosis	8	192	0.017937474
	**Down**	dre00100	Steroid biosynthesis	6	22	8.02 × 10^−8^
		dre01100	Metabolic pathways	36	1719	6.16 × 10^−6^
		dre00510	N-glycan biosynthesis	4	59	0.003553768
		dre00591	Linoleic acid metabolism	3	29	0.003544349
		dre04672	Intestinal immune network for IgA production	3	37	0.007089709
**CuTri**	**Up**	dre00830	Retinol metabolism	14	80	6.54 × 10^−9^
		dre01100	Metabolic pathways	75	1719	1.90 × 10^−8^
		dre00982	Drug metabolism—cytochrome P450	9	59	1.19 × 10^−5^
		dre00591	Linoleic acid metabolism	6	29	6.12 × 10^−5^
		dre00350	Tyrosine metabolism	6	38	0.000295976
	**Down**	dre00564	Glycerophospholipid metabolism	7	107	0.001203711
		dre00561	Glycerolipid metabolism	5	69	0.003967923
		dre04110	Cell cycle	7	150	0.007960662
		dre04114	Oocyte meiosis	7	146	0.006895526
		dre00062	Fatty acid elongation	3	42	0.026082223

**Table 3 ijms-24-11203-t003:** Primer sequences.

Genes	Name	Primer Sequences
*jak2a*	F R	TAATGATGAGAGAACA GATCATCCACAGTTCAG
*apoeb*	F R	CTCTTGTGGTATTCTTTGCTCTGGCAGTTT TTGCACCATGCCGTCAGTTTGTGTGTTGAG
*Per1a*	F R	GTACATCTCATCCCAGGCGG AAAGGTGCTGACGTCCTGAG
*clcn2c*	F R	GAGACCTGAGGTGTGTGAGC TAACACACATTGGTCGCGGA
*caps3a*	F R	GATCGCAGGACAGGCATGAA AATCTGCGCAACTGTCTGGT
*pck1*	F R	CGCGTACTGGAGTGGATGTT GTGTGTTGCGTGTCTTCAGC
*egr1*	F R	AAGTGAGATCAGCCTGGTGC GGTTGCAAAGGCCTTGATGG
*star*	F R	ACCTGTTTTCTGGCTGGGATG GGGTCCATTCTCAGCCCTTAC
*il7r*	F R	TGCAAAGAGCAAAGGAGGGC CAACATAATCCACGCGGTGT
*actb*	F R	GTCCGTGACATCAAAGAG ACCGCAAGATTCCATAC
*ef1α*	F R	AGGAAGATTGAACGCAAGAGTG AGGAAGATTGAACGCAAGAGTG

## Data Availability

The data are available upon reasonable request.

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
