# Peer review of "The Combined Use of Copper Sulfate and Trichlorfon Exerts Stronger Toxicity on the Liver of Zebrafish"

_ijms, 2023, doi:10.3390/ijms241311203_

Round 1

Reviewer 1 Report (Previous Reviewer 3)

The authors of “Different Toxic Effects of Copper Sulfate and Trichlorfon Alone or in Combination on Zebrafish Liver” analyzed the toxicity of these two compounds on zebrafish.  The authors argue that since these compounds are used extensively in aquaculture, examining them in a zebrafish model will aid in out understanding of the limitations of these compounds in aquaculture. Overall, this is an interesting incremental advance in the study of these commonly used aquaculture additives and demonstrate that zebrafish can serve as a model system for their toxicity.  I made 8 specific requests for improvements in the previous review.  The authors have addressed all of my concerns and I recommend that the paper be accepted.  

Author Response

Thank you very much for your support and assistance throughout this research. We truly appreciate it! Thank you!

Reviewer 2 Report (Previous Reviewer 2)

The authors have improved the article including new data. However the explanation of these results is again too vague or incorrect:

- Line 108-109. "a significant increase in endoplasmic reticulum proliferation" I assume you mean expansion of the ER

- Line 109-110. "The degradation of internal hepatocyte structures..." What do you mean by degradation? What structures?

- Line 110. again degradation...

- Line 113. Could the authors explain what a mitochondrial vacuole is?

- At least in my version, figure 1 has no figure legend. And Fig. 1J is not even mentioned in the text

The expansion of the ER, the increase number of vacuoles, together with the increased lysosomes that the authors do not described, might suggest that autophagy is trigger to compensate toxicity. Does de molecular data give some information respect to this?

Author Response

Thank you very much for your valuable suggestions regarding this research. In response to your questions, we have provided answers that include images. The modified portions in the article are highlighted with a yellow background. To ensure clarity, we have uploaded the specific answers in PDF format. We hope this meets your requirements. Thank you!

Round 2

Reviewer 2 Report (Previous Reviewer 2)

Thanks for the response

This manuscript is a resubmission of an earlier submission. The following is a list of the peer review reports and author responses from that submission.

Round 1

Reviewer 1 Report

This manuscript tried to compare the individual and simultaneous effects of copper sulfate and trichlorfon on liver of zebrafish, but the purpose of the paper is not clearly expressed in the paper, and the explanation of the results seems to be insufficient. A simple list of results is not enough to reach a conclusion, and explanations and understanding of each figure do not seem sufficient. Therefore, it seems difficult to recommend publication.

 This manuscript explains about different effects of copper sulfate and trichlorfon alone or in combination on zebrafish liver. Results showed that combination treatment were more toxic. But explanation about toxic effect of combination treatment were insufficient. So, I recommend you to modify according to the following comments.

Title

  1. Please write the title more clearly. It would be better you explain toxic effects more specific in title

Abstract

  1. In line 27, Liver damage is ambiguous. Please indicate in detail.

Introduction

  1. In line 53~54, You wrote ‘However, excessive Cu can not only kill pathogenic microorganisms, but also cause damage to fish,~’. But, it is little confusing so I think ‘However, excessive Cu not only can kill~ but also ~’ is better.
  2. In line 74, I recommend you to explain more detail about why Cu and Tri are often used together to achieve the best effects of fish disease prevention.
  3. Please explain the abbreviation of the CAT, SOD, AKP, GPX, GO and DEG in Introduction not Results and Methods

Results

  1. I recommend you explain background of antioxidant and immune enzyme
  2. I recommend you explain the purpose of GO enrichment analysis and KEGG enrichment analysis

Discussion

  1. You mentioned liver is a crucial detoxification and immune organ in fish but, I recommend you should mention the reason why you target the liver as a main target organ.

Figure

  1. In figure 1, Does not designated the A is what and B is what.
  2. In figure 2, we don’t know what kind of staining experiment it is. So please write figure legend properly.
  3. In figure 2, I recommend you do analysis the staining result and make a graph indicating the damage value to see effect of the combination treatment.
  4. In figure 2, It will be better to display where the vacuolar areas and the scale of the figure.
  5. In figure 2, If you want to conclude like in line 230, please indicate the P value between copper sulfate, tri and combination.
  6. In figure 3, The bar identification is not quite specific.
  7. In figure 3, Please indicate which sample the significance is compared to.

Moderate Editing for English is required. 

Author Response

Comment (C) 1: Title 1. Please write the title more clearly. It would be better you explain toxic effects more specific in title.

Response (R)1: Thanks for your comments. We have changed the title as “The Combination Use of Copper Sulfate and Trichlorfon Exert Stronger Toxicity on the Liver of Zebrafish” in the revised manuscript.

Comment (C) 2: Abstract 1. In line 27, Liver damage is ambiguous. Please indicate in detail.

Response (R)2: Thanks for your comments, we have changed this description as “hepatocyte structure damage” in the revised manuscript.

Comment (C) 3: Introduction 1. In line 53~54, You wrote ‘However, excessive Cu can not only kill pathogenic microorganisms, but also cause damage to fish,~’. But, it is little confusing so I think ‘However, excessive Cu not only can kill~ but also ~’ is better.

Response (R)3: Thanks for your comments, we have modified this in the revised manuscript.

Comment (C) 4: Introduction 2. In line 74, I recommend you to explain more detail about why Cu and Tri are often used together to achieve the best effects of fish disease prevention.

Response (R) 4: Thanks for your comments, we have added more detail about why Cu and Tri are often used together to achieve the best effects of fish disease prevention as “Copper sulfate is a widely used disinfectant in aquaculture, effectively inhibiting the growth of bacteria, fungi, and planktonic organisms, thus reducing the spread of pathogens. Trichlorfon, also known as an organophosphate insecticide, is a broad-spectrum pesticide that effectively controls pests such as mosquitoes, flies, mites, and copepods in aquaculture systems. The combined use of copper sulfate and trichlorfon allows for simultaneous disinfection and pest control in aquaculture, thereby enhancing the effectiveness of disease and pest management.”.

Comment (C) 5: Introduction 3. Please explain the abbreviation of the CAT, SOD, AKP, GPX, GO and DEG in Introduction not Results and Methods.

Response (R) 5: Thanks for your comments, we have explained the abbreviation of CAT, SOD, AKP and GPX in the revised manuscript as “The antioxidant system plays a crucial role in combating oxidative stress and maintaining the redox balance within cells. This system includes important enzymes such as superoxide dismutase (SOD), catalase (CAT), alkaline phosphatase (AKP), and glutathione peroxidase (GPX). These enzymes work together to neutralize reactive oxygen species (ROS) and protect cells from oxidative damage. These components of the antioxidant system contribute to cellular health and play a vital role in various physiological processes.”.

Comment (C) 6: Results 1. I recommend you explain background of antioxidant and immune enzyme

Response (R) 6: Thanks for your comments, we have explained background of antioxidant and immune enzyme in the introduction of revised manuscript from line 64 to line 70.

Comment (C) 7: Results 2. I recommend you explain the purpose of GO enrichment analysis and KEGG enrichment analysis.

Response (R) 7: Thanks for your comments. GO analysis allows for the annotation of a large number of genes into different functional categories. Through GO analysis, we can gain insights into gene sets that are associated with specific biological processes, cellular components, or molecular functions within a gene expression dataset. This helps to uncover the functions and interactions of genes in specific biological processes. On the other hand, KEGG analysis maps gene expression data to specific pathways, revealing the roles of genes in cellular metabolism and signal transduction. This aids in understanding the interactions and regulatory mechanisms of genes in specific biological processes.

Comment (C) 8: Discussion 1. You mentioned liver is a crucial detoxification and immune organ in fish but, I recommend you should mention the reason why you target the liver as a main target organ.

Response (R) 8: Thanks for your suggestions, we have mentioned the reasons as “The liver plays a crucial role in detoxification, lipid metabolism, nutrient storage, and immune function in fish. Selecting the liver as the main target organ for research helps to gain in-depth understanding of various aspects of fish physiology, health, and adaptability. ” in the revised manuscript from line 24 to line 27.

Comment (C) 9: 

Figure 1. In figure 1, Does not designated the A is what and B is what.

  1. In figure 2, we don’t know what kind of staining experiment it is. So please write figure legend properly.
  2. In figure 2, I recommend you do analysis the staining result and make a graph indicating the damage value to see effect of the combination treatment.
  3. In figure 2, It will be better to display where the vacuolar areas and the scale of the figure.
  4. In figure 2, If you want to conclude like in line 230, please indicate the P value between copper sulfate, tri and combination.
  5. In figure 3, The bar identification is not quite specific.
  6. In figure 3, Please indicate which sample the significance is compared to.

Response (R) 9: Thanks for your suggestions. We have made modifications to these images based on the suggestions you provided in the revised manuscript.

Reviewer 2 Report

In this article the authors analyze the damage produced by Cu and Tri in zebrafish liver. Both are commonly used as disinfectants and insecticides. Thus, addressing the effects of these compounds is of interest to the aquaculture field. The authors unravel several genes affected by these treatments through transcriptomics. While this part is quite robust, the histology one, a crucial part to really understand liver damage, is weak. Further experiments to assess liver damage must be performed. This might include cell death, accumulation of fat, electron microscopy…

I think this article could be interesting. However, it provides little novel data. Cu treatment produce liver damage, as well as Tri treatment. Certainly, combination of both at the same concentration is going to be worse. It would be more interesting if the authors, using their analysis, would have found a combined treatment that keeps its properties but does not damage liver, or at least it does to a lower extent.

Specific comments:

-        Please include a lower magnification and then high magnification in the histology. Please include scale bars.

-        Line 85-86. The hepatocyte structure was significantly damage. How? The description is vague and many of the details cannot be observed in the images provided. To really understand the changes in the hepatocyte morphology, electron microscopy is necessary.

-        Line 86-87. Intercellular spaces are visibly enlarged and is comparable between Tri and Cu. Quantification or scoring should be included.

-        Line 87. Vacuolar areas. I am not that sure that these are vacuoles. Liver damage can lead to increase lisosomal activity and accumulation of lipid droplets. All these organelles stain similarly in H&E. Again, if “vacuolar areas” are increased, please quantify.

-        Line 89. A significant reduction of hepatocytes. Please quantify. Are there pyknotic nuclei? Are there any kind of necrosis? Semithin sections stained with toluidine blue could be a nice addition.

-        Line 91. Enucleated cell membranes. I am not sure what the authors mean with this. But, there is no way you can observe this process with H&E

-        Line 114-117. I assume this refers to Fig. 5? Reference to figure 5 is missing. In general, formatting must be improved. Results start in figure 2. Figure 1 is much farther in the methods section. Conclusions are after methods.

-        Discussion is too long, and it reads like a repetition of results. For example, the third paragraph only includes a general article about AKP. However, there is no mention to previous articles that have studied the immune response in this scenario.

-        Sentences such as line 183-184, I think, can be eliminated. Again, in that paragraph authors mixed cell cycle and immune response and repeat their CAT, SOD and GPX results.

-        Line 194-198. At 21 days CAT, SOD, GPX were significantly downregulated. An alternative hypothesis would be that hepatocytes are dead and there are no mitochondria. Further experiments addressing mitochondrial function should be included.

-        Paragraph 5 talks about immune response, again, and then metabolism. If the authors are interested in the immune response, further experiments should be included. Macrophage response can be easily studied in zebrafish liver.

-        Please, include the reference number of the kits used to quantify enzymatic activity or explain it further.

-        This is just a curiosity. If hepatic tissue was extracted, why so many genes related to meiosis are affected?

Author Response

Comment (C) 1: Please include a lower magnification and then high magnification in the histology. Please include scale bars.

Response (R)1: Thanks for your comments. We have added the scale bars in the revised manuscript.

C2: Line 85-86. The hepatocyte structure was significantly damage. How? The description is vague and many of the details cannot be observed in the images provided. To really understand the changes in the hepatocyte morphology, electron microscopy is necessary.

C3: Line 86-87. Intercellular spaces are visibly enlarged and is comparable between Tri and Cu. Quantification or scoring should be included.

C4: Line 87. Vacuolar areas. I am not that sure that these are vacuoles. Liver damage can lead to increase lisosomal activity and accumulation of lipid droplets. All these organelles stain similarly in H&E. Again, if “vacuolar areas” are increased, please quantify.

C5: Line 89. A significant reduction of hepatocytes. Please quantify. Are there pyknotic nuclei? Are there any kind of necrosis? Semithin sections stained with toluidine blue could be a nice addition.

C6: Line 91. Enucleated cell membranes. I am not sure what the authors mean with this. But, there is no way you can observe this process with H&E.

R2-6: Thank you for your comments. We have addressed the questions related to liver tissue sections uniformly. Thanks for your useful suggestions. We apologize for the previous inaccurate description, and in the revised article, we have provided a new description. However, since we are not proficient in measuring the vacuolar area of hepatic tissue sections, we instead quantified the relative quantity of hepatocyte per unit section area and performed a significance test. The relevant results are presented in Section 2.1 Morphology observation and Fig. S1.

C7: Line 114-117. I assume this refers to Fig. 5? Reference to figure 5 is missing. In general, formatting must be improved. Results start in figure 2. Figure 1 is much farther in the methods section. Conclusions are after methods.

R7: Thank you for your comments. We have added the citation to Fig. 5 in the revised

manuscript. We have also rearranged the order of the figures in the revised manuscript.

C8: Discussion is too long, and it reads like a repetition of results. For example, the third paragraph only includes a general article about AKP. However, there is no mention to previous articles that have studied the immune response in this scenario.

R8: Thank you for your comments. We have revised and reduced the description of the results section, while incorporating additional references regarding AKP in the revised manuscript.

C9: Sentences such as line 183-184, I think, can be eliminated. Again, in that paragraph authors mixed cell cycle and immune response and repeat their CAT, SOD and GPX results.

R9: Thanks for your comments. We have eliminated this sentence in the revised manuscript.

C10: Line 194-198. At 21 days CAT, SOD, GPX were significantly downregulated. An alternative hypothesis would be that hepatocytes are dead and there are no mitochondria. Further experiments addressing mitochondrial function should be included.

R10: Thanks for your comments. We agree with your viewpoint that there is indeed a possibility of hepatocyte death, and we are currently conducting a study that explores this aspect.

C11: Paragraph 5 talks about immune response, again, and then metabolism. If the authors are interested in the immune response, further experiments should be included. Macrophage response can be easily studied in zebrafish liver.

R11: Thanks for your comments. In the present study, immune damage was only one aspect of investigating the overall toxicity of copper and trichlorfon on zebrafish. We did not extensively delve into this area, but future research can specifically examine their impact on zebrafish immune function.

C12: Please, include the reference number of the kits used to quantify enzymatic activity or explain it further.

R12: Thanks for your comments. We have added the specific information of the relevant kit in the revised manuscript.

C13: This is just a curiosity. If hepatic tissue was extracted, why so many genes related to meiosis are affected?

R13: Thanks for your comments. In normal circumstances, meiosis is a cell division process specific to reproductive cells, which is responsible for generating gametes (sperm and eggs). The liver is typically not involved in meiosis, as its primary functions include metabolizing substances, detoxification, and maintaining homeostasis in the body. However, in certain diseases or abnormal conditions, gene expression in the liver can be disrupted, including genes related to meiosis. This condition may be associated with abnormalities in the cell cycle, disturbances in cell differentiation, or tumor formation, among other factors. Although the exact role of meiosis-related genes in the liver may not be clear, the abnormal gene expression in these cases may provide important clues about the pathophysiology of the liver and warrant further investigation.

Reviewer 3 Report

The authors of “Different Toxic Effects of Copper Sulfate and Trichlorfon Alone or in Combination on Zebrafish Liver” analyzed the toxicity of these two compounds on zebrafish.  The authors argue that since these compounds are used extensively in aquaculture, examining them in a zebrafish model will aid in out understanding of the limitations of these compounds in aquaculture. 

The authors first verified the concentrations of Cu and trichlorfon in there water samples using ICP-MS as shown in figure 1.  In figure 2 the authors then performed HE staining on liver sections, and observed reduced nuclei and enlarged intercellular spaces indicative of liver damage.  In figure 3 the authors observed increased expression of CAT, SOD, and GPX 7 days after treatment.  After 24 days levels dropped below baseline.  The authors then engaged in transcriptome analysis with these agents.  In figure 4 they show Venn diagrams for Copper treatment, Tri treatment or a combination.  In figure 5 they verify their RNA-seq data by comparing to RT-qPCR analysis for 9 genes and observed significant consistency between the two techniques.  Table 2 shows the top significantly enriched GO terms and table 3 shows the most significantly enriched KEGG pathways, while figure 6 shows a heatmap of up and down regulated genes. 

Overall, this is an interesting incremental advance in the study of these commonly used aquaculture additives and demonstrate that zebrafish can serve as a model system for their toxicity.  After addressing a few issues, this manuscript would be suitable for publication in IJMS.

Major comments:

1. It is confusing for the read to have figure 1 referred to only in the materials and methods, while figures 2-6 are mentioned earlier in the paper.  I would recommend adding an introductory paragraph to the results discussing why it is important to directly verify the Cu and Tri concentrations and refer to figure 1. 

2.  Another issue throughout the paper is the figure legends have too little information in each.  For example “first” and “second” are not defined in the legend for figure 1, only in the materials and methods.  It would be easier for the reader if these were defined in the legend.  Also, how the data was collected and how the error bars were generated should be mentioned here and not just in the Materials and Methods.  In figure 2 it would be helpful to reiterate that these are cryosections stained with H and E.  It would also be helpful to add some labels to make it easier for the reader to identify nuclei and intercellular gaps.  In the legend for figure 3 I would recommend including the full names for CAT, SOD, GPX and AKP. 

3.  I would also like a more thorough discussion of the observation in figure 3 that CAT, SOD and GPX are upregulated at 7 days but down regulated at 21.  The authors argue that the 7 day time point was indicative of an acute stress response and by the 21 day time-point they claim “the body’s antioxidant capacity continued to weaken, suggesting that the damage caused by copper sulfate and Tri … had exceeded the repair threshold of the body itself”.  This is a major claim in the paper that needs more justification.  Are there referenced that demonstrate a similar trend after toxin treatment in zebrafish?  Do the authors have any evidence of increased cell damage at these later stages? 

Minor comments:

In line 63 it is unclear what “short physical impacts” means

In line 96 I would recommend defining CAT, SOD, GPX and AKP. 

In line 152 (and other places) the authors use the phrase “various life activities” which is a vague term and should be clarified.  Also, the authors use the phrase “of the body” or “in the body” repeatedly throughout the discussion.  It often adds little to the sentence and can be somewhat distracting. 

In line 201 the authors state  “resulting in significant down-regulation of transcriptome levels during reproduction”.  This phrase is confusing, and I assume they mean that reproductive genes are down regulated in expression. 

In line 240, why were the fish maintained at 25 C?  Was this for convenience or for a specific experimental reason. 

Author Response

C1: It is confusing for the read to have figure 1 referred to only in the materials and methods, while figures 2-6 are mentioned earlier in the paper. I would recommend adding an introductory paragraph to the results discussing why it is important to directly verify the Cu and Tri concentrations and refer to figure 1.

R1: Thanks for your suggestions. We have made adjustments to the order of the figures in the revised manuscript.

C2: Another issue throughout the paper is the figure legends have too little information in each.  For example “first” and “second” are not defined in the legend for figure 1, only in the materials and methods.  It would be easier for the reader if these were defined in the legend.  Also, how the data was collected and how the error bars were generated should be mentioned here and not just in the Materials and Methods.  In figure 2 it would be helpful to reiterate that these are cryosections stained with H and E.  It would also be helpful to add some labels to make it easier for the reader to identify nuclei and intercellular gaps.  In the legend for figure 3 I would recommend including the full names for CAT, SOD, GPX and AKP.

R2: Thanks for your suggestions. We fully accept your suggestions and have added some key information to the figure legends in the revised manuscript.

C3: I would also like a more thorough discussion of the observation in figure 3 that CAT, SOD and GPX are upregulated at 7 days but down regulated at 21.  The authors argue that the 7 day time point was indicative of an acute stress response and by the 21 day time-point they claim “the body’s antioxidant capacity continued to weaken, suggesting that the damage caused by copper sulfate and Tri … had exceeded the repair threshold of the body itself”.  This is a major claim in the paper that needs more justification.  Are there referenced that demonstrate a similar trend after toxin treatment in zebrafish?  Do the authors have any evidence of increased cell damage at these later stages?

R3: Thanks for your comments. In the present study, we observed that under the combined exposure of copper sulfate and trichlorfon, the antioxidant enzyme activities in the fish were up-regulated as a response to oxidative stress. However, as the exposure time prolonged, the antioxidant enzyme system in the fish was damaged, leading to the occurrence of oxidative stress. In fact, the liver tissue sections clearly showed an increase in hepatocyte damage in the later stages of exposure.

C4: In line 63 it is unclear what “short physical impacts” means

R4: Thanks for your comments. The original reference indicated that the “short physical impacts” encompassed hematological and blood biochemical indicators.    

C5: In line 96 I would recommend defining CAT, SOD, GPX and AKP.

R5: Thanks for your comments. We have defined CAT, SOD, GPX and APK in the revised manuscript from line 66 to line 72.

C6: In line 152 (and other places) the authors use the phrase “various life activities” which is a vague term and should be clarified.  Also, the authors use the phrase “of the body” or “in the body” repeatedly throughout the discussion.  It often adds little to the sentence and can be somewhat distracting.

R6: Thanks for your suggestions. We agree with your point of view and have modified the phrase “various life activities” to “physiological activities” accordingly.

C7: In line 201 the authors state “resulting in significant down-regulation of transcriptome levels during reproduction”.  This phrase is confusing, and I assume they mean that reproductive genes are down regulated in expression.

R7: Thanks for your comments. We have revised the inaccurate description in the revised manuscript accordingly.

C8: In line 240, why were the fish maintained at 25 C?  Was this for convenience or for a specific experimental reason.

R8: Thanks for your comments. In the present study, we maintained the water temperature at 25°C to create a stable environment and prevent any stress-related impacts on the experimental results due to temperature fluctuations. It was not done for any specific experimental purpose but rather for convenience and consistency.

Round 2

Reviewer 2 Report

Thanks for your reply. Please include the scale bars, I cannot see them in my version.

Author Response

Thanks you very much for your comments. We have reuploaded the Figure 1 with the scale bars in the revised manuscript.